



# How to account for irrigation withdrawals in a watershed model

Elisabeth Brochet[1], Sabine Sauvage[2], Youen Grusson[1], Ludovic Lhuissier[3], and Valérie Demarez[1]

[1]Centre d'Etudes Spatiales de la Biosphère, Université de Toulouse, CNES/CNRS/IRD/INRA/UPS, 18 av. Edouard Belin, bpi 2801, CEDEX 9 31401 Toulouse, France
[2]Centre de Recherche sur la Biodiversité et l'Environnement, ENSAT, CNRS/UPS/INPT, Av. de l'Agrobiopole, bpi 32607, 31326 Castanet Tolosan CEDEX, France
[3]Compagnie d'Aménagement de Côteaux de Gascogne, Chemin de Lalette, CS 50449, 65004 Tarbes CEDEX, France

**Correspondence:** Valérie Demarez (valerie.demarez@univ-tlse3.fr) and Sabine Sauvage (sabine.sauvage@univ-tlse3.fr)

**Abstract.** In agricultural areas, the downstream flow can be highly influenced by human activities during low flow periods, especially dam releases and irrigation withdrawals. Irrigation is indeed the major use of freshwater in the world. This study aims at precisely taking these factors into account in a watershed model. The Soil and Water Assessment Tool (SWAT+) agro-hydrological model was chosen for its capacity to model crop dynamics and management. Two different crop models

were compared in their ability to estimate water needs and actual irrigation. The first crop model is based on air temperature as the main determining factor for the growth, whereas the second relies on high resolution data from Sentinel-2 satellite to monitor plant growth. Both are applied at plot scale in a watershed of $800 \, \text{km}^2$ characterized by irrigation withdrawals. Results show that including remote sensing data leads to more realistic modeled emergence dates for summer crops. However both approaches have proven to be able to reproduce the evolution of daily irrigation withdrawals throughout the year. As a result,

both approaches allowed to simulate the downstream flow with a good daily accuracy, especially during low flow periods.

## 1 Introduction

The water cycle is substantially affected by climate change. The scientific consortium World Weather Attribution have shown that the 2022 climatic and agricultural drought in Europe would have been about 5 times less likely to happen in the 1900s, and the frequency of such drought will increase further in the coming years. This 2022 drought event has been triggered not only

by heatwaves but also by a lack of soil water storage (Schumacher et al., 2022). Many regions suffered from water scarcity, inducing some conflicts between users. Water demand for drinking water and irrigation are in addition expected to increase due to population growth. Water management during low flow period will therefore be one of the major challenges in future years. Agriculture currently account for about 70% of withdrawals in the world (UNESCO, 2015). In this context, it is important to better assess water needs for agriculture. Efficient tools are then needed to better know the cultivation practices, including

irrigated areas, crop dynamics and irrigation practices. These parameters must be assessed on a spatial extents relevant for water management, such as river basin or sub-basin scale.

Watershed models are already used by multiple actors, at multiple time scales and spatial extends. Combined with climate change scenarii, they allow a better understanding of the upcoming challenges for the next decades. For instance, Ex-



plore2070 project (Carroget et al., 2017) have shown with high confidence that the summer stream flow in France in years
2050-2070 will be on average 30 to 70% lower than during years 1990-2010. Combined with management scenarii, watershed models can support decision making for agricultural policies (Murgue et al., 2014; Allain et al., 2018). Furthermore, dam managers and local authorities can make use of short term forecast to optimize the low flow management (E-tiage software https://www.e-tiage.com/; PREMHYCE platform https://webgr.inrae.fr/projets/projets-en-cours/onema-premhyce/; Nicolle et al., 2020). Hence the need for continuous improvement of these models.

However, most studies at watershed scale don't precisely include human activities, especially withdrawals for irrigation. For instance, most users of SWAT (Soil and Water Assessment Tool) model don't explicitly include withdrawals in their studies (Fohrer et al., 2014; Boithias et al., 2014; Martin et al., 2016; Cakir et al., 2020). In such studies, dam releases and withdrawals are indirectly taken into account through the calibration performed to fit the downstream flow. As a result, the performance of low flow simulation is quite low, as highlighted by Boithias et al. (2014).

Some modelers include cultivation practices through default crop management or through past years' statistics, but these models have shown limitations for the prediction of irrigation. Leenhardt and Lemaire (2002) indeed describe a case for which the sowing of summer crops was delayed due to the unusually wet spring, but the model didn't account for this and predicted the maximum irrigation demand one month too early. In the same way, Senthilkumar et al. (2015) highlight the impact of cultivar earliness on their water needs. Models that fit better the actual crops dynamics and management are therefore needed. Remote
sensing (RS) offers a real added value for spatial and temporal variability. Since 2016, thanks to the Copernicus program, optical satellite images (Sentinel-2) with high spatial (10 m) and temporal (five days) resolution are available anywhere in the world. These resolutions are appropriate to get information on crops dynamics and cultivation practices at plot scale, over a whole watershed.

Methods to retrieve the cultivation practices from RS data has been developed for more than twenty years (Guérif and Duke,
1998; Launay and Guerif, 2005), and has bloomed since short revisit time satellites were launched (Courault et al., 2010; Ferrant et al., 2014). The Sentinel constellation, offering a high spatial resolution and high revisit time, has made this task easier and more accurate (Pageot et al., 2020 for irrigated areas; Bazzi et al., 2021 for irrigation events; Rolle et al., 2022 for sowing dates).

For the purpose of water managing, RS can also be combined with crop water requirements models. The FAO-56 method
(Allen et al., 1998) is the most widely used land-surface model, in which the potential evapotranspiration (ET) is adjusted with a crop coefficient and a stress coefficient. The classical method uses standard values of crop coefficient, that implies standard growth conditions, but an increasing number of studies replaces them with remote sensing data (Saadi et al., 2015; Etchanchu et al., 2017; Battude et al., 2017; Yousaf et al., 2021; Kharrou et al., 2021; Maguire et al., 2022), in order to account for the current year conditions. In particular, Etchanchu et al. (2017) highlight the contribution of remote sensing data at plot scale,
that makes more robust calculation of LAI (leaf area index) and ET in comparison to the classical method. In addition to ET calculation, these methods can be used to estimate irrigation, through a soil water balance (Saadi et al., 2015; Battude et al., 2017; Kharrou et al., 2021; Maguire et al., 2022) or a comparison with a reference crop coefficient (Yousaf et al., 2021). Even



if the retrieval of irrigation suffers from large uncertainties at farm scale (Saadi et al., 2015 for instance), Olivera-Guerra et al. (2023) have shown that proper calibration can greatly reduce the uncertainties at irrigation district scale.

Despite these recent advances in the modeling of crop dynamics and water needs, very few works investigate the effect of those cultivation practices on stream flow, at high spatial and temporal scales. The MAELIA platform (Modeling of socio-Agro-Ecological system for Landscape Integrated Assessment, Murgue et al., 2014) appears as an exception since it combines the SWAT hydrological module with an agent-based model to perform dams operation and crop management. In this case, modelers have inferred large set of decision rules either from surveys (Leenhardt et al., 2004; Maton et al., 2005, 2007; Murgue et al.,

2014) or from a farm advisor (Clavel et al., 2011). The authors were able to check the relevance of the annual irrigation amount predicted by the model, but not the timing of irrigation due to the lack of daily data. The present study intents to go further than the MAELIA project in terms of spatialization, by substituting part of the decision rules by remote sensing estimates. Indeed, feeding the model with observations could improve its robustness, while avoiding the complexity of an agent-based model.

Some studies already aimed to assess the benefits of optical remote sensing in agro-hydrological models, without focusing

on crop irrigation. They intended to improve the ET component of the water balance by forcing (Martin et al., 2016; Paul et al., 2021; Jin et al., 2022, all in SWAT model) or assimilating (Kumar et al., 2019 in Noah-MP model; Mohammadi Igder et al., 2022 in SWAT+ model) the RS LAI (remote sensing leaf area index) in the model and short-cut the plant growth algorithm. Under temperate or arid climate types, ET indeed appears to be the main component of the water balance, hence a small relative error on ET could lead to a larger relative error on the river discharge. However, even if these authors report an improvement on

yield or carbon fluxes, they report little modification on the stream flow, ranging from 2 to 7%. Explanations for these results could be that the modeled LAI, even if uncertain, is enough to compute ET, or that the uncertainty on other hydrological or anthropogenic fluxes are larger than the one on ET. As Paul et al. (2021) and Jin et al. (2022) both performed calibration with and without remote sensing data, it also raises the question : can calibration correct the errors in ET values, in a way that it hides the possible benefits of RS data ? Moreover, as far as we know, no studies at watershed scale used remote sensing data at

plot scale, even though Sentinel data would be suitable for that.

The objective of this study is to take into account human activity during low flow period in the modeling process of an agricultural watershed. Unlike previous studies, our work is located on a watershed where irrigation have a huge impact on stream flow: in summer, the downstream flow can be 4 times lower than the upstream flow, due to withdrawals. To reach this objective, crop models at plot scale are used to predict crop water requirements and irrigation withdrawals and the SWAT+

model is used to compute water stocks and fluxes. The native crop growth module of SWAT is compared to a new module that uses Sentinel-2 high resolution data to compute ET. An innovative calibration method for hydrological fluxes is introduced to address several calibration issues: the short time depth of Sentinel-2 data, the fact that calibration can hide the possible benefits of RS data, and the highly influenced stream flow. Numerous data sources have been used to carry out the calibration of irrigation trigger threshold, the estimation of irrigation sources, and the assessment of daily withdrawals. Finally, the relevance

of such model, combining plot scale and watershed scale during the low flow period, is assessed.



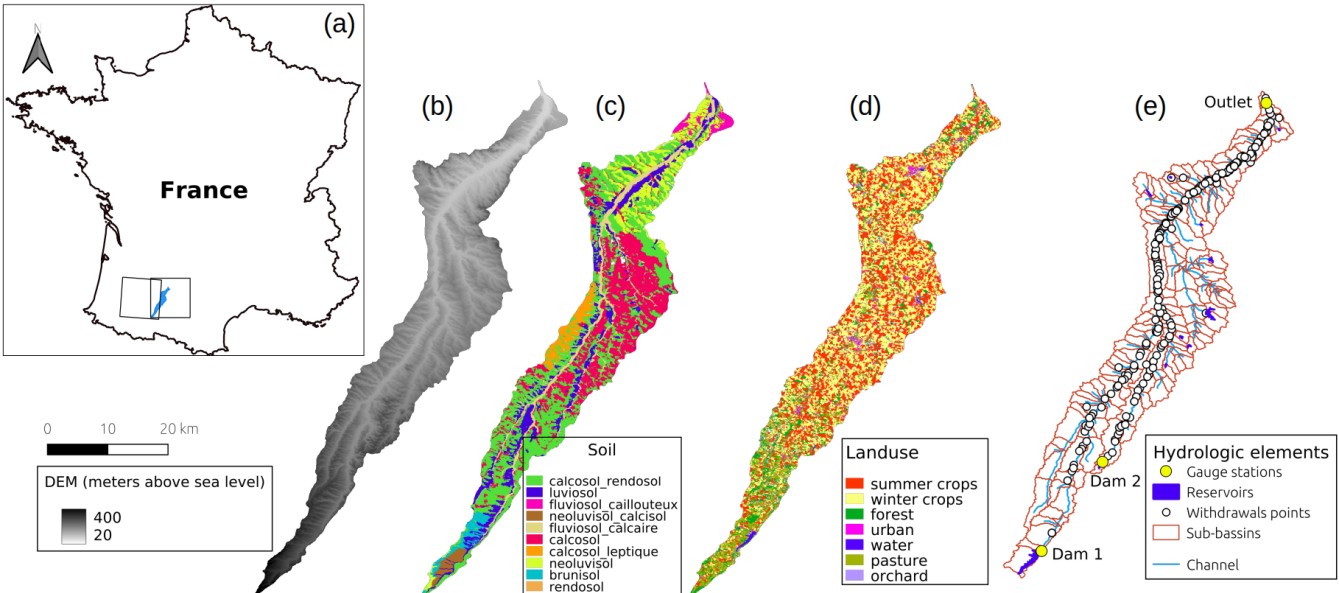

**Figure 1.** Maps of the study area. (a) Location of Gimone watershed in South-West France, as well as Sentinel-2 footprints; (b) Digital elevation model (1" resolution); (c) Soil map (1/250.000 resolution); (d) Crop pattern for year 2017 (plot scale); (e) Hydrological network setup of the model.

## 2 Materials and methods

### 2.1 Study area

The Gimone watershed (800 $km^2$) has been selected for this study. It is located in southwestern France, in a temperate climate zone. (Fig. 1). The average rainfall is 700 mm. The flow rate at outlet ranges from 0.4 to 40 $m^3 s^{-1}$, with an average of 3 $m^3 s^{-1}$. Agricultural land is dominant (85%) in this watershed. Irrigated crops (mainly corn, soybean, orchards and vegetables) represent about 10% to 15% of the total area, and irrigation water is withdrawn from rivers and from individual reservoirs. Other crops, mainly winter wheat, winter rapeseed and sunflower, are not irrigated. Due to the high water demand and low water availability in summer, two dams were built upstream in the 1990s, on the Gimone and Marcaoué rivers, with capacities of respectively 24 $Mm^3$ and 1.5 $Mm^3$ (Fig. 1-e). The biggest dam is filled with water diverted from the Neste river. Both dams are managed by the CACG (*Compagnie d'Aménagement des Côteaux de Gascogne*) and their purpose is to maintain a minimal flow rate in rivers during low flow periods (DOE - *Débit Objectif d'Etiage* – minimum target flow of 0.4 $m^3 s^{-1}$), and limit the frequency of irrigation restrictions. As a result, the stream flow during irrigation period is strongly influenced by dam releases and irrigation withdrawals, and has little to do with hydrological processes.



## 2.2 Data overview

The topography of the watershed derives from the SRTM (Shuttle Radar Topography Mission) digital elevation model at 1" resolution (Fig. 1-b). Daily weather data - precipitation, air temperature, wind velocity, solar radiation and air humidity - were extracted from the SAFRAN product (Durand et al., 1993). It is an interpolation based on weather measurements and model reanalysis, on a 8 km resolution grid. Soil properties, namely the depth, bulk density and clay and sand content, were found in the french soil map RRP (*Référentiel Régional Pédologique*, regional reference document for pedology) at 1/250.000 resolution (Fig. 1-c); the maximal root depths was set according to Rigou (2016); the available soil water content (AWC) was derived from Bruand et al. (2003) pedotransfert functions; the initial estimate of the saturated hydraulic conductivity comes from measurements. Land use data come from the French database of agricultural land use (RPG – *Registre Parcellaire Graphique*) from years 2015 to 2021, completed by the OSO map (Inglada et al., 2017) for non-agricultural areas (Fig. 1-d). Stream flow coming out of the two dams was provided by the CACG.

In addition to the "classical" data to set up the SWAT+ model, high resolution remote sensing data were used. Since 2017, the two Sentinel-2 satellites provide images every five days, with resolution ranging from 10 to 60 m depending on the wavelength. The level 2A images from two Sentinel-2 tiles, 31TCJ and 30TYP (Fig. 1-a) are used in this study. They are post-processed through the MAJA chain (Hagolle et al., 2015) that corrects atmospheric effects and produces a cloud mask. These clouds masks are used in the interpolation process, to obtain gap-filled time series.

River discharge at the outlet was also provided by the CACG, and has been used for calibration and validation. In order to get an overview of the origin of withdrawals, the PAR (*Plan Annuel de Répartition* – Annual plan for water allocation) database was used. It gathers individual withdrawal permissions issued every year by the French administration, and contains the authorized volumes in each type of water resource. The PKGC (*Pratiques Culturales en Grandes Cultures* – Agricultural practices in field crops) database, carried out every five years, consists on a large survey that covers all French departments with information about 28000 agricultural plots. Dataset from 2017 has been used in this study, in particular the sowing dates and amounts of annual irrigation of 160 corn and soybean fields in southwestern France.

A large part of the irrigation pumps of the Gimone watershed are equipped with networked meters (Fig. 1-e). It provides daily measurement of water withdrawn from the two main rivers, the same two that are regulated with dams and for which low flows are sustained during summer (Gimone and Marcaoué). Not all intake points are equipped for the years considered in the present study: 33% of the pumps were equipped in 2019, 62% in 2020 and 72% in 2021. For this reason, only 2020 and 2021 data were used, and corrected with a proportionality coefficient to convert those data into total withdrawals.

## 2.3 SWAT+ model

### 2.3.1 Global overview

The Soil and Water Assessment Tool (Arnold et al., 1993, 2012) is a agro-hydrological semi-distributed and process-based model. To setup this semi-distributed approach, the simulated watershed is divided into multiple sub-basins according to topography. Each sub-basin is divided into HRU (hydrological response units), which are homogeneous areas in terms of



slope, soil and land use. Calculation of the process-based water balance is then carried out at HRUs level on a daily basis (Nietsch et al., 2001). Daily rainfall is split between surface runoff and infiltration using the curve number method (Soil Conservation Service, 1972; Rallison and Miller, 1982). Soil water sustains evapotranspiration and leads to subsurface runoff
and percolation into aquifers. If the simulated water table of the shallow aquifer is high enough, ground water flow from the aquifer supports river flow. This contribution of aquifers to the river flow is called baseflow. Land use management (e.g. planting, irrigation, fertilization, harvest) occurs on the basis of user-defined decision rules. When irrigation is applied, the origin of withdrawals (river, aquifer, reservoir) must be specified so that the amount can be subtracted from the appropriate source. All fluxes reaching the rivers (surface and subsurface runoff, ground water flow) are aggregated at the sub-basin level
and routed through the stream network to the next sub-basin downstream, until the watershed outlet.

### 2.3.2 Model setup

The 60.5.3 version of SWAT+ has been used in this study as the baseline model, from which features and options were added (see Code availability Section).

The watershed has been divided into 149 sub-basins (Fig. 1). In this study, we forced HRUs to match individual fields in
all agricultural land, so that using remotely sensed indices at plot scale would be consistent. This delineation resulted in about 26000 HRUs. The model was fed with the true crop rotation from 2015 to 2021 according to the French database RPG.

### 2.4 Two approaches for crop growth and management

Two methods have been compared to simulate crops seasonal dynamics. The first one is the native crop growth module of SWAT, based on a simplified version of EPIC (Erosion Productivity Impact Calculator). The second one consists on obser-
vation of measured crop growth through the NDVI, a remote-sensing-based index linked to vegetation development. The two approaches are hereafter called "SWAT-O" and "SWAT-NDVI" respectively. The SWAT-O setup has also been run without any irrigation, to assess the impact of withdrawals on the stream flow. Below is a short description of both methods.

### 2.4.1 SWAT-O : Growth of plants and ET driven by heat units (EPIC)

Heat Units (namely the average temperature of the day, minus the base temperature) drive the growth of LAI (leaf area index)
according to the following formula (Barnard, 1948; Phillips, 1950):

$$fr_{LAImx} = \frac{fr_{PHU}}{fr_{PHU} + \exp\left(l_1 - l_2 \cdot fr_{PHU}\right)} \tag{1}$$

$$fr_{PHU} = \frac{\sum_{i=1}^{d} HU_i}{PHU} \tag{2}$$

where $fr_{LAImx}$ is the fraction of maximum LAI, $l_1$ and $l_2$ are shape coefficients, $HU_i$ is the heat units of day $i$ (°C) and $PHU$ is the potential heat units required for plant maturity (°C). More detailed description of this simplified EPIC module can
be found into the SWAT theoretical documentation (Nietsch et al., 2001).



**Table 1.** Decision rules for corn emergence : The emergence occurs on the first day that all the conditions are met. YHU : Total heat units of the year; SWC : soil water content; FC : field capacity.

| SWAT-O | SWAT-NDVI |
|---|---|
| $HU_0 > 30\%$ YHU | April 20th < day < July 10th |
| $SWC < 1.05 \cdot FC$ | NDVI increasing |
|  | $0.2 < NDVI < 0.5$ |

The beginning of growth is settled according to decision rules, where the most important parameter is the sum of heat units since January 1st ($HU_0$). The emergence occurs on the first day that all the conditions are met. See example for corn in Table 1.

Potential evapotranspiration (PET) of the crop depends on its LAI based on Penman-Monteith equation (Monteith, 1965) :

$$\lambda E_t = \frac{\Delta(H_{net} - G) + \gamma K_1 \cdot (0.622\lambda \cdot \frac{\rho_{air}}{P}) \cdot \frac{e_z^o - e_z}{r_a}}{\Delta + \gamma \cdot (1 + r_c/r_a)} \tag{3}$$

$$r_c = \frac{r_l}{0.5 \cdot LAI} \tag{4}$$

where $r_c$ is the stomatal resistance ($\mathrm{s\,m^{-1}}$), $r_l$ the stomatal resistance ($\mathrm{s\,m^{-1}}$) of a single leaf, $E_t$ is the maximum transpiration ($\mathrm{mm\,d^{-1}}$), $\lambda$ is the latent heat of vaporization ($\mathrm{MJ\,kg^{-1}}$), $\Delta$ is the slope of the saturation vapor pressure-temperature curve ($\mathrm{kPa\,°C^{-1}}$), $H_{net}$ is the net radiation ($\mathrm{MJ\,m^{-2}\,d^{-1}}$), $G$ is the heat flux density to the ground ($\mathrm{MJ\,m^{-2}\,d^{-1}}$), $\gamma$ is the psychometric constant ($\mathrm{kPa\,°C^{-1}}$), $\rho_{air}$ is the air density ($\mathrm{kg\,m^3}$), $P$ is the air pressure ($\mathrm{kPa}$), $K_1$ is a dimension coefficient (K1=86400 $\mathrm{s\,d^{-1}}$), $e_z^o$ is the saturation vapor pressure ($\mathrm{kPa}$), $e_z$ is the vapor pressure of air ($\mathrm{kPa}$), and $r_a$ is the aerodynamic resistance ($\mathrm{s\,m^{-1}}$).

Actual evapotranspiration is adjusted from this PET through a stress coefficient calculated as

$$K_s = \begin{cases} \exp\left(5\left(\frac{SWC}{0.25 \cdot Zr \cdot AWC} - 1\right)\right) & \text{if } \frac{SWC}{0.25 \cdot Zr \cdot AWC} < 1 \\ 1 & \text{else.} \end{cases} \tag{5}$$

where SWC is the soil water content ($\mathrm{mm}$) and $Zr$ is the root depth ($\mathrm{mm}$).

### 2.4.2 SWAT-NDVI : Growth of plants and ET driven by optical remote sensing

A Normalized Difference Vegetation Index (NDVI) is calculated from Red and Near-Infrared (NIR) bands (4th and 8th bands) of Sentinel-2 and is used as a proxy for vegetation growth:

$$NDVI = \frac{NIR - Red}{NIR + Red} \tag{6}$$



Bare soil NDVI is lower than 0.2 whereas crop NDVI ranges from 0.2 at early stages to 0.7-0.9 at full development. Table 1
shows rules used to detect crop emergence with NDVI.

Actual evapotranspiration ($E_t$, mm) is calculated with FAO-56 dual coefficient method (Allen et al., 1998)):

$$E_t = (K_{cb} \cdot K_s + K_e) \cdot ET_0 \tag{7}$$

with $ET_0$ being the evaporative demand (mm), $K_{cb}$ the crop basal coefficient, $K_e$ the soil evaporative coefficient, and $K_s$
the stress coefficient. $K_{cb}$ can be seen as a linear function of NDVI, of which the coefficients were fixed following Toureiro
et al. (2017):

$$K_{cb} = 1.464 \cdot NDVI - 0.253 \tag{8}$$

The $K_e$ factor depends on the crop coefficient $K_{cb}$, the fraction of soil coverage $Fcover$, and a reduction coefficient $K_r$:

$$K_e = \min\left(K_r \cdot (1.2 - K_{cb}); 1.2 \cdot (1 - Fcover)\right) \tag{9}$$

$$Fcover = 1.23 \cdot NDVI - 0.15 \qquad \text{(Saadi et al., 2015)} \tag{10}$$

and $K_r$ depends on the soil texture and SWC, according to Merlin et al. (2011).

The stress coefficient is calculated as

$$K_s = \frac{SWC}{(1 - p) \cdot Zr \cdot AWC} \tag{11}$$

where $p$ is the depletion coefficient that is usually equal to 0.55 in the FAO-56.

## 2.5 Model for irrigation

Based on local water managers knowledge, all fields with a slope lower than 10% and covered by corn, soybean, orchard, or
vegetables were considered as irrigated. The actual source of irrigation withdrawals (river or reservoirs) is not known. The
PAR database however provides information about the distribution of authorized withdrawals between sources. We assumed
real withdrawals distribution to be the same. It results in 58% of the water pumped from the rivers, whereas 42% comes
from small reservoirs (36% from connected reservoirs and 6% from disconnected ones). As small reservoirs (<10 ha) are not
included in our setup, irrigation water was considered to come from the closest river from each HRU. This should be kept in
mind for the interpretation of results.

Rules are required to trigger the simulated irrigation. In order to implement them, we used several surveys conducted in
southwestern France to identify farmers' behaviors (Leenhardt et al., 2004; Maton et al., 2005, 2007; Senthilkumar et al.,





**Table 2.** Decision rules for irrigation of corn. SWC : soil water content; $fr_{PHU}$ : fraction of YHU.

| type of rule | **SWAT-O** | **SWAT-NDVI** |
|---|---|---|
| water stress | $SWC < (1 - p_{trig}) \cdot Zr \cdot AWC$ | |
| weather forecast | $\text{rain}(t) < 10\,\text{mm}$ and $\text{rain}(t+1) < 20\,\text{mm}$ | |
| minimum time between two irrigation | 10 days | |
| irrigation depth (starter) | If $0.1 < fr_{PHU} < 0.2$ : $15\,\text{mm}$ | |
| irrigation depth (normal) | If $0.2 < fr_{PHU}$ : $30\,\text{mm}$ | |
| stop irrigation (senescence) | $fr_{PHU} < 0.9$ | If NDVI is decreasing : NDVI $> 0.95 \cdot \max (\text{NDVI})$ |

**Table 3.** Parameters selected for calibration as well as the range within which they can vary.

| Name | Description | Range | Unit |
|---|---|---|---|
| $Ksat$ | saturated hydraulic conductivity | $0.4Ksat - 1.5Ksat$ | $\text{mm}\,\text{d}^{-1}$ |
| $Cn_2$ | curve number for partition of rain between surface runoff and infiltration | $0.9Cn_2 - 1.1Cn_2$ | - |
| $perco$ | ability of soil water to reach aquifer | $0.5 - 1$ | - |
| $latq\_co$ | adjustment coefficient for lateral subsurface flow | $0 - 1$ | - |
| $alpha\_bf$ | groundwater transfert time to the river | $0.005 - 0.04$ | $\text{d}^{-1}$ |

2015). Table 2 presents the implemented rules which include soil water content, the rainfall of the day and the next day as well
as crop phenological stages.

The $p_{trig}$ parameter (Table 2) can vary depending on the crop, and was calibrated for corn and soybean. The calibration
method is similar to the one described by Olivera-Guerra et al. (2023). Calibration and validation data are the annual irrigation
depth for 160 irrigated fields in the PKGC database. This led to a value of 0.57 for corn and soybean, and 0.55 for silage corn.

## 2.6 Calibration of hydrological processes in SWAT

Stream flow in the Gimone river during summer is more impacted by human activities (release of water from dams, with-
drawals) than by natural hydrological processes. In order to take this into account, calibration of "natural" hydrological param-
eters has been performed on months where anthropogenic impact is reduced: from November to May.

The calibration of parameters related to "natural" hydrology has been performed only on the SWAT-O setup and calibrated
values were also used for the SWAT-NDVI setup for several reasons. First, Sentinel-2 remote sensing data are only available
since 2017. The SWAT-NDVI setup would have not allow to cover a sufficiently long period to perform calibration and valida-
tion over contrasted hydrological years. This choice has also been made to make a more straightforward comparison between
the two crop models. Indeed, if calibration had been performed on both setups, the specificity of each crop model could have
been hidden.



**Table 4.** Four best parameter sets according to $NSElog$.

| $Ksat$ (rel.) | $Cn_2$ (rel.) | $perco$ | $latq\_co$ | $alpha\_bf$ |
|:---:|:---:|:---:|:---:|:---:|
| 0.78 | 0.96 | 0.55 | 0.12 | 0.0057 |
| 0.75 | 0.97 | 0.53 | 0.20 | 0.0061 |
| 1.06 | 0.91 | 0.66 | 0.16 | 0.0071 |
| 0.65 | 0.92 | 0.86 | 0.36 | 0.0067 |

**Table 5.** Calibration and validation scores.

| | $NSEsqrt$ | $NSElog$ | $NSE$ | $KGEsqrt$ | $R^2$ | $Pbias$ |
|:---|:---:|:---:|:---:|:---:|:---:|:---:|
| **Calibration** (2017-2021) | 0.70 | 0.78 | 0.55 | 0.84 | 0.74 | 5.7% |
| **Validation** (2012-2016) | 0.65 | 0.70 | 0.44 | 0.82 | 0.69 | 11.6% |

Years 2017 to 2021 have been used as calibration period, and 2012-2016 as validation period. Years 2010 and 2011 have
been run as a warm-up period.

Sensitive parameters have been identified through a sensitivity analysis using a one-at-a-time procedure as described by
Abbaspour (2015) and based on previous study conducted in this watershed (Grusson et al., 2015; Cakir et al., 2020). This
allowed to adjust the range of each parameter before starting the calibration process. The parameters are listed in Table 3.

Calibration has been performed by randomly drawing 1000 sets of parameters following a uniform distribution within tested
range of each parameter. The model was then run for each of these 1000 sets over the 2010-2021 period. For each of the
1000 runs, several metrics have been calculated on daily flow over the 5-years calibration period, excluding June to October.
Metrics are the Nash-Sutcliffe efficiency (Nash and Sutcliffe, 1970) performed on the logarithm ($NSElog$) and the square root
($NSEsqrt$) of the discharge, as well as the Kling-Gupta efficiency on the square root of the discharge ($KGEsqrt$) (Gupta
et al., 2009). The square root ($sqrt$) and logarithm ($log$) transforms respectively put the emphasis on middle-range and low
flows (Pushpalatha et al., 2012; Santos et al., 2018). For each score independently, the top 1% parameter sets were analyzed
through box plots. The dispersion between those best sets allows to highlight the equifinality issue during calibration process.

Goodness of fit on validation period was then evaluated with the same metrics, in addition with $R^2$ and $Pbias$. Eventually,
$NSElog$ only was used to select the parameters values, because the calibration months (November to May) still contains low
flows, but very little influenced by withdrawals. In order to assess the uncertainty due to calibration, the four best parameter
sets according to $NSElog$ were selected for the rest of this study.





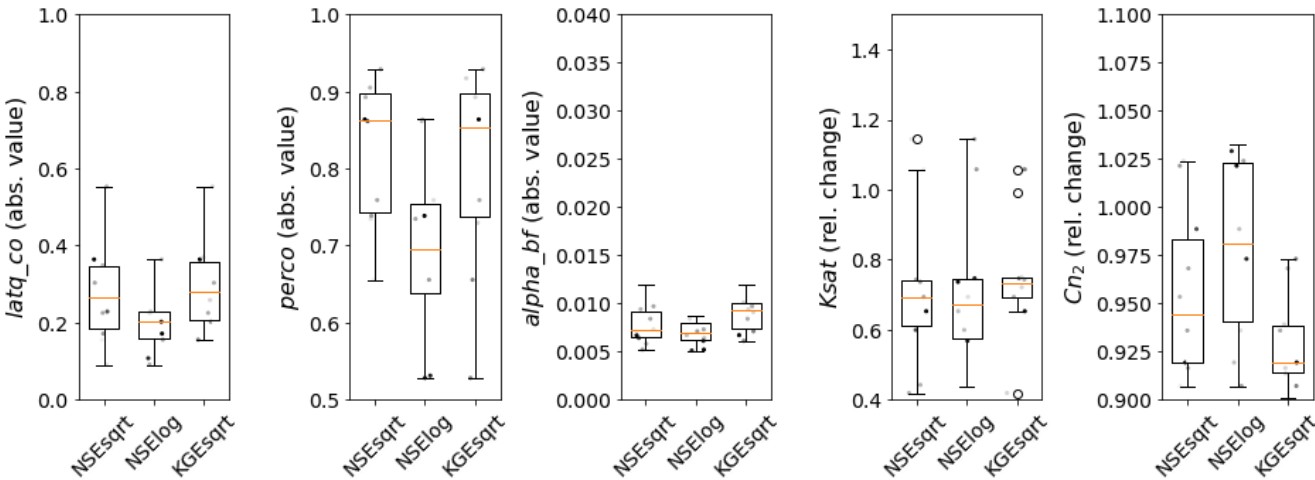

**Figure 2.** Box plot analysis of the parameter values of the the 10 best simulations according to three different metrics.

**Table 6.** Annual water balance from the calibrated SWAT model, averaged over years 2012-2021, and stream flow separation.

| **In** (mm) | rainfall | | 695 | |
|---|---|---|---|---|
| | input from Neste channel | | 24 | |
| | | surface runoff | | 8% |
| | outlet | sub-surface runoff | | 64% |
| | flow | baseflow | 126 | 14% |
| **Out** (mm) | | Neste channel | | 13% |
| | water diverted to Save river | | 7 | |
| | evapotranspiration | | 578 | |
| | deep percolation | | 2 | |
| **Change in** | soil water content | | 6 | |
| **storage** (mm) | rivers+reservoirs+aquifers | | <0.5 | |

# 3 Results and discussion

## 3.1 Calibration and hydrology outside the low water period

Figure 2 shows the dispersion of the top 1% sets of parameters according to each criteria independently ($NSEsqrt$, $NSElog$, $KGEsqrt$). Since the bounds of Y axis correspond to the range of each parameter, a low dispersion of values reveals a high

sensitivity of the parameter, and a low equifinality.

Accordingly, $alpha\_bf$ and to a lesser extent $Cn_2$ are sensible parameters. A relative value of $Cn_2$ below 1 means that the runoff over infiltration ratio must be lower than the default values. A value of $alpha\_bf$ around 0.005 to 0.01 d$^{-1}$ means that





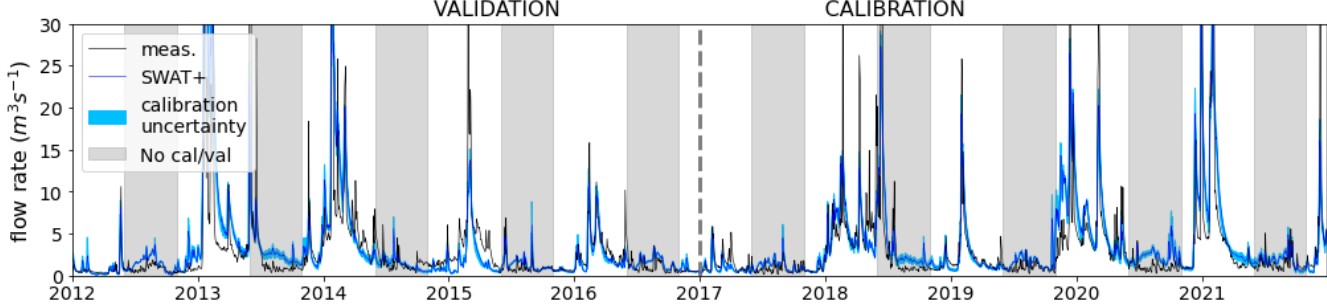

**Figure 3.** Daily measured and simulated outlet stream flow. Grey areas highlight the periods removed from the calibration-validation process.

the transfer time from aquifer to river is about 3 to 6 months. Although no transfer time measurements are available to check the credibility of this value, the simulated base flow is consistent with the base flow index (BFI) of the river calculated with

Wallingford method (Hamon, 1963; ESPERE software from BRGM https://www.brgm.fr/fr/logiciel/espere-estimation-pluie-efficace-recharge-selon-differentes-methodes, Lanini et al., 2020).

On the contrary, $latq\_co$ and $Ksat$ values seem scattered over a wider range. But further analysis have shown that these parameters are highly correlated, because they are both used to calculate the sub-surface lateral flow : if $latq\_co$ is low then $Ksat$ must be high and vice versa. Hence the conclusion that when the value of one of these two coefficients is fixed, then the

other becomes very sensitive. Only $perco$ coefficient seems to have a lower sensitivity.

Due to the dispersion of the best parameters without large difference in the scores, it seemed appropriate to keep more than one set of parameters. This allows to account for calibration uncertainty in future runs. The four bests sets of parameters according to $NSElog$ are therefore kept (Table 4). Table 5 shows the values of metric on calibration (2017-2021) and validation (2012–2016) periods for the simulation using the first parameter set. It is worth noting that these scores are calculated only

over the months of November to May.

The $NSE$ value is only satisfactory (Moriasi et al., 2007), but since $NSE$ mostly assesses the performance for high flows, which are not the goal of this study, and knowing the limitation of discharge measurement for high flows, these values are seen as sufficient here. On the other hand, the $NSElog$ values and the $KGEsqrt$ values are very good in calibration period and good in validation period, according to Moriasi et al. (2007, 2015) classification. That indicates a good agreement for medium

and low simulated flow rates compared to measurements. As for $R^2$ and $Pbias$, these values are considered as good and very good respectively.

The annual water balance averaged over 10 years (2012-2021) simulated from the calibrated values is detailed in Table 6. Rainfall (around 700 mm per year) is mostly converted into evapotranspiration (around 580 mm per year) and flow (126 mm per year). In comparison, the measured annual flow is 116 mm per year. In this section of the study, the simulated stream

flow is not yet reduced by withdrawals, which can explain most part of the difference. This water balance is very similar to the one obtained by Boithias et al. (2014) on a close and similar watershed. By not including human influence on the stream flow, they reported a way lower performance in low flow periods compared to high flow periods. This is the reason





why our hydrological calibration is performed only on the November-May period, whereas the influenced flow (June-October) performance is investigated in the following sections. Table 6 also shows that the two-thirds of the river flow comes from subsurface runoff. Baseflow accounts for about 14% of the total flow.

The daily calibrated stream flow at the outlet is shown in Fig. 3. The four simulations with the four selected sets of parameters are plotted as a minimum-maximum colored range, along with a plain line for the median value.

## 3.2 Plants phenology

The emergence of crops follows the rules described in Section 2.4. As a result, the emergence dates have very different distributions depending on the method. In the SWAT-O setup, most of the crop growths are synchronized. For instance, in 2017, the model grows all corn during the last week of May. The reason is that the growth in SWAT-O is mostly determined by air temperature, which is nearly homogeneous over the watershed. On the other hand, the use of NDVI to detect the beginning of growth spreads the emergence dates over almost two months. Figure 4 shows this effect on the 700 corn fields in the Gimone watershed. To decide which distribution is the more realistic, those dates have been compared to 150 corn fields from the PKGC database located within 80 km of the study area. The assumption is made that the dynamics of these plots and of Gimone plots are similar, due to the similarity of altitudes, latitudes and climate. The PKGC database contains the sowing date, whereas the use of NDVI allows the detection the phenological stage of 6-8 leaves (here called "emergence" for more clarity). To be consistent between NDVI and PKGC estimations, a fixed amount of 300 heat units has been added to the database sowing dates to converted it into plausible emergence dates, $300\,°\mathrm{C}\,\mathrm{d}^{-1}$ being the average need to reach the 6-8 leaves stage from sowing.

Comparison between the PKGC database and the models clearly shows that spread emergence dates (SWAT-NDVI) fits better to real dates. Surveys conducted in an other sub-watershed in South-Western France in 2005 also showed a spread of sowing dates over five weeks (Maton et al., 2007). However the distribution of emergence dates in SWAT-NDVI does not perfectly match the PKGC distribution either. The major explanation might be the lack of cloud-free Sentinel-2 images during months April to June. In 2017, only four dates were usable : April 6[th], May 16[th] and 26[th], June 25[th]. As images were not available for 40 days, the linear interpolation may lead to unrealistic NDVI time series. In addition, decision rules leads to some artifacts. The peak on April 30[th] is produced by a lot of plots that already fulfilled the condition on NDVI, and that reached the condition on soil water content on the same day in the model. Even if the retrieved emergence dates in the SWAT-NDVI mode were closer to reality, the PKGC date could also not be a representative sample of the diversity of plots. As a consequence, the above comparison focuses on the range of emergence dates without a quantitative description of the distributions.

## 3.3 Evapotranspiration

Figure 5 shows the monthly simulated ET for years 2017 to 2021. Both setups lead to quite similar ET, with a monthly average difference of 5% and a daily average difference of 17% at watershed scale. On average over the five years, ET simulated with SWAT-NDVI setup is slightly lower than in the SWAT-O setup, by 1.5%.

These low differences are in line with the finding of Martin et al. (2016), who also tested the influence of emergence dates retrieved with RS on the water balance. They reported a change in ET from 0 to ±5% depending on the months. Their biggest



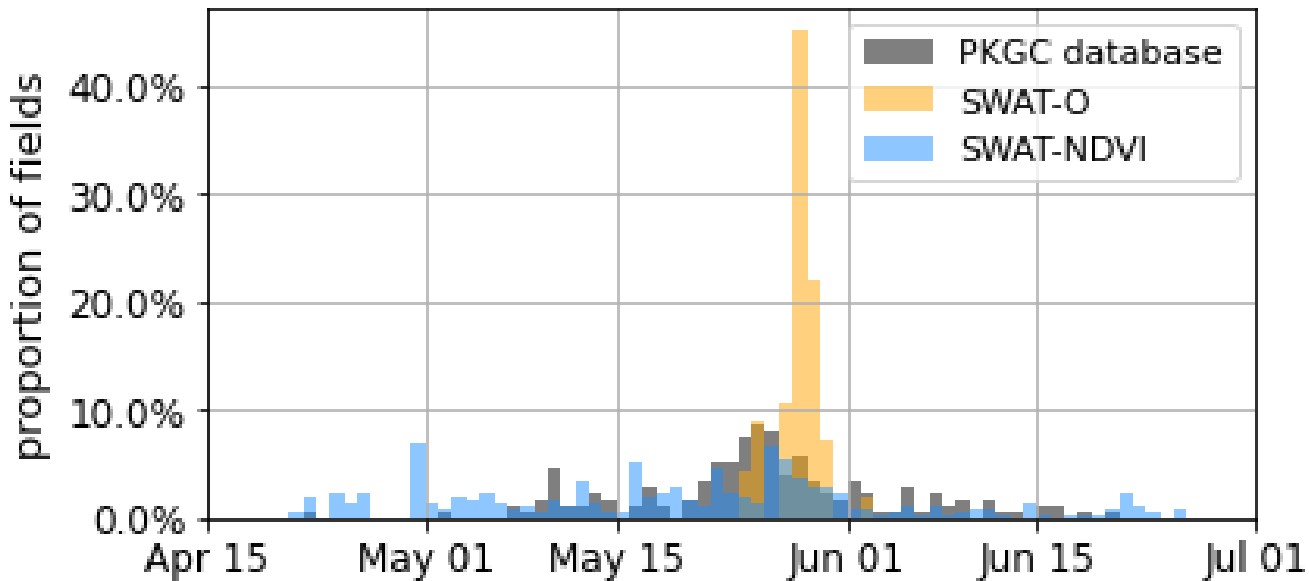

**Figure 4.** Distribution of corn emergence dates for year 2017. Comparison between SWAT-O and SWAT-NDVI modes, against the reconstructed emergence dates from PKGC database.

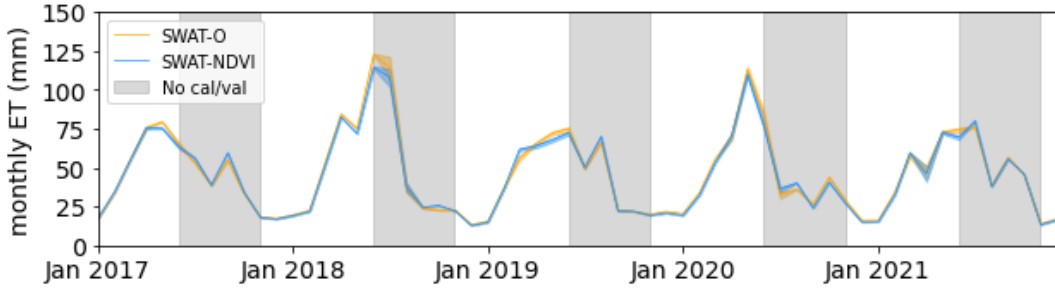

**Figure 5.** Monthly evapotranspiration in both setups.

monthly change (around 10%) occurred on one year where the actual growth of crop was far ahead of the theoretical growth calendar. In the present study however, no year stands out in particular.

### 3.4 Daily withdrawals

Figure 6 provides a daily comparison between the simulated water withdrawals and those that are measured by the networked
water meters.

It can be seen that the timing and amounts of withdrawals is highly depends on the year. The weather might therefore be the first determining factor for irrigation. The rainfall in June and July 2021 resulted in low evaporative demand and very low





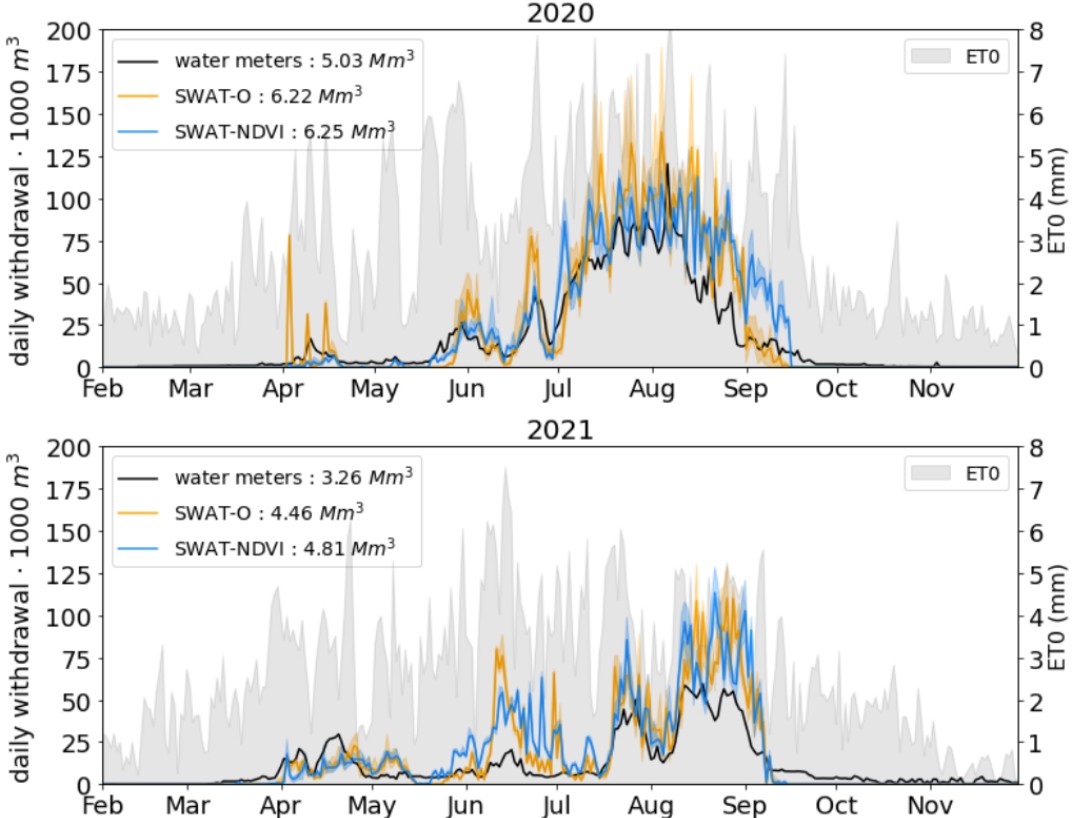

**Figure 6.** Simulated and measured daily irrigation withdrawals in the two managed rivers (Gimone and Marcaoué), for years 2020 and 2021.

irrigation during this period. On the contrary, high temperatures and a lack of rain in July 2020 resulted in a high evaporative demand, followed by large amounts of irrigation.

Some differences can be also seen between simulated withdrawals in SWAT-O and SWAT-NDVI modes. In particular, the peak of water demand in SWAT-NDVI mode at the beginning of the irrigation season (June) is reduced in intensity and spread in time compared to the SWAT-O mode. This is a direct effect of the spreading of emergence dates, that triggers a spread in crop management dates.

At the end of the 2020 irrigation season (September), SWAT-NDVI mode produced a higher irrigation amount than SWAT-O
mode. This is due to a difference in crop phenology between the two setups: the NDVI showed that corn and soybean were not yet harvested and that their water needs were still high due to the high $ET_0$. On the contrary, in SWAT-O mode, most crops were already harvested.

The remaining gaps between the simulations and measurements could be explained by several factors:

(1) The actual irrigated fields are not known. Our choice to irrigate all corn, soybean, vegetable and orchard on fields with
slope lesser than 10% is consistent but arbitrary. As several authors (Demarez et al., 2019; Pageot et al., 2020; Puy et al., 2022)





**Table 7.** Scores for the SWAT-O and SWAT-NDVI setups, for years 2017 to 2021. Up: "High flow" period, from November to May. Down: "Low flow" period, from June to October.

| High flow : November to May | | | | | | |
|---|---|---|---|---|---|---|
| | $NSEsqrt$ | $NSElog$ | $NSE$ | $KGEsqrt$ | $R^2$ | $Pbias$ |
| **SWAT-O** | 0.70 | 0.78 | 0.55 | 0.84 | 0.74 | 5.7% |
| **SWAT-NDVI** | 0.70 | 0.79 | 0.55 | 0.82 | 0.74 | 11.2% |

| Low flow : June to October | | | | | | |
|---|---|---|---|---|---|---|
| | $NSEsqrt$ | $NSElog$ | $NSE$ | $KGEsqrt$ | $R^2$ | $Pbias$ |
| **SWAT-O** | 0.63 | -0.07 | 0.60 | 0.78 | 0.79 | 1.1% |
| **SWAT-NDVI** | 0.63 | 0.37 | 0.61 | 0.78 | 0.79 | 10.5% |

highlight, irrigated areas are the main driver of water withdrawals. A 10% error in areas indeed logically leads to a 10% error in withdrawals.

(2) This graph compare the withdrawals that occur only in the two main rivers. We assumed that it represents 56% of the total withdrawals, but the actual distribution of withdrawals origin might be slightly different.

(3) Intrinsic uncertainty on the networked meters is very low, but errors could originate from the fact that only 60% to 75% of the intake point are equipped, and that we extrapolated these withdrawals data to the missing intake points.

(4) Water restrictions are frequent at the end of the low water season but are not taken into account in the model. The overestimation in September 2020 can indeed be explained by the drought decree on August 29[th], 2020, that ordered a 50% reduction of withdrawals in the main rivers and prohibited irrigation from small tributaries. In contrast, 2021 was a wet year

and main rivers has not been subjected to any restriction.

(5) April to June being a rainy period, Sentinel-2 images are often corrupted by clouds. During this period, summer crop are sown and begin to grow. For this reason, the linear interpolation between clouds-free images fail to reproduce the convex shape of the NDVI and leads to an overestimation, which results in an overestimation of the ET. Therefore, irrigation can also be overestimated during this period.

**3.5 Effects of irrigation withdrawals on the river flow**

In addition to the SWAT-O and SWAT-NDVI setups, a third setup was run, SWAT-O with no irrigation. Each setup was run 4 times with different sets of parameters (see Calibration section). In the following, the "error bars" or "uncertainty margins" refer to the range of values obtain with those four runs.

For further quantitative analysis, we distinguish between two periods. On the one hand, the "high flow" period from Novem-

ber to May, over which the calibration of hydrological processes have been performed. On the other hand, the administrative "low flow" period, from June to October, where the flow is highly influenced by dams releases and irrigation withdrawals.





**Figure 7.** Simulated against measured daily outlet stream flow during the June-October period, for five consecutive years. The SWAT-O and SWAT-NDVI setups are compared with the theoretical simulation without irrigation withdrawals.

During the high flow season, most of the metrics are not influenced by the introduction of NDVI method into the model (Table 7-Up). The $Pbias$ values rises from 5.7% to 11.2%, which remains into acceptable values (Moriasi et al., 2007). Since





hydrological processes were calibrated without NDVI, a slight decrease in model performance could have been expected after
adding it.

On the other hand, Table 7-Down shows the metrics calculated over the low flow season. The scores over this period barely changed, but the significant increase of $NSElog$ from about 0 to close to 0.4 suggests an improvement of low flow simulation by the SWAT-NDVI mode.

Figure 7 shows the simulated stream flow at the outlet for the period where Sentinel-2 data are available (2017 to 2021). The
graph focuses on the irrigation period, from April to October, where the differences between setups are more likely to be seen, and this is the period of interest regarding the water demand and the strategies for water management.

The graph reflects as far as possible the uncertainties we are aware of. For simulation plots, plain lines are the median value, and the errors bars are minimum and maximum values obtain from the four runs with different parameters sets. In addition, there might be a $\pm10\%$ error on the measured flow rate values, according to the water manager . This uncertainty is included
on the graph through a colored range.

The minimal flow rate that has to be maintained (DOE) is also included as a horizontal line.

Significant differences can be observed between the three setups. On the one hand, both models fail to simulate the flow rate at the end of low flow season for 2018, 2019 and 2020, for which heavy rains occurred in October. The reason for this might be that a lot of small reservoirs – not taken into consideration in this study – are close to be empty at the end of summer, and
are filled up by the first heavy rains period (bar plots in Fig. 7), leading to less flow reaching the river.

On the other hand, the comparison with the no-irrigation mode clearly shows the importance of taking in account agricultural practices, through withdrawals here, that accounts for more than half of the stream flow during summer in such watershed. It shows that the order of magnitude of simulated withdrawals is correct, which is a second validation of the retrieved withdrawals.

Finally, the improved timing of withdrawals in the SWAT-NDVI setup has some visible effects on the stream flow mostly
at the beginning of irrigation season (June, July). The flow is smoother in this mode, because it is subject to fewer peaks in withdrawals. In general, the SWAT-NDVI mode produces a higher discharge than the SWAT-O mode. This is due to the lower ET in SWAT-NDVI mode compared to SWAT-O mode (Fig. 5) and the lower amount of withdrawals (Fig. 6).

## 4   Conclusions and outlooks

This study deals with the modeling of crop phenology and irrigation withdrawals in an agricultural zone, and their impact on
downstream flows. The SWAT+ agro-hydrological model was chosen because it allows a spatialized and detailed description of crop management. Two modeling approaches were compared for crop management. The first modeling approache (SWAT-O) uses heat units to compute the emergence and senescence dates of crops, their growth rate and LAI, and then their ET. The second approach (SWAT-NDVI) uses Sentinel-2 remote sensing data, through the NDVI vegetation index, to determine all these features.

Before tackling the crop modeling, the calibration of hydrological fluxes was addressed. Because of the very influenced stream flow during summer, calibration was performed on the remaining months, from November to May. This uncommon



calibration method relies on the assumption that parameters are independent of the time of the year, and this hypothesis has proven to be plausible in the course of this study. A strength of this study is to admit the uncertainties due to calibration and take it into account in the analysis.

Then, focusing on crops, the SWAT-NDVI setup first allows to better account for the spatial and temporal variability of the cultivated crops, by spreading the emergence over time and over the watershed, which is more realistic. The modeled ET in this new setup follows the observed crop dynamics at plot scale, even though the differences in ET seem barely significant at watershed scale. The simulated daily withdrawals have been compared against daily irrigation data provided by the water manager. Both setups reproduce quite well the overall evolution of withdrawals throughout the year. However, the SWAT-NDVI

mode shows the best performances at the beginning of the irrigation season, without further calibration. This improvement can mainly be attributed to the spreading of crop management dates.

In our watershed, irrigation withdrawals and dam releases are the main determining factors for the river flow in summer. Since dam releases are known and irrigation withdrawals are simulated, the model is able to provide satisfactory daily stream flows in low flow periods. The results suggest that the SWAT-NDVI mode allows a little better accuracy for very low flows, but

this would need to be confirmed by other experiments.

To conclude, this study shows that it is possible to simulate a very influenced stream flow with modeling tools. Indeed, very few modelers could focus on daily flow during low flow periods, due to the difficulties in modeling both hydrology and anthropogenic processes. In this study, we assessed two methods for crop dynamics, that allows to retrieve withdrawals and flow dynamics during irrigation period, with an unusual accuracy.

It would of course be interesting to reproduce this experiment in different watersheds with different climate and agricultural practices. In areas where land use, crop dynamics and irrigated areas are not well known, remote sensing methods could be of great help. In this study, the hypothesis about irrigated areas appeared to be sufficient. However in other watersheds where less plots are irrigated, maps of irrigated areas could be necessary (Pageot et al., 2020). In addition, in order to obtain more accurate emergence dates, radar images, that are not affected by clouds could be used in combination with optical images (Rolle et al.,

405    2022).

In most watersheds, the lack of knowledge about the origin of withdrawals still remains a limitation of our approach. Finally, a major unknown of many watersheds are the small reservoirs, that may have a huge impact on the flow rate, but about which very few information is available (Boisson et al., 2022).

*Code availability.* The custom SWAT+ sources are available at https://github.com/ElisabethJustin/SWATplus-NDVI

*Author contributions.* E.B., V.D. and S.S. designed the experiments; E.B. conducted the experiments; E.B., V.D., Y.G. and S.S. analysed the results; L.L. provided the data and gave expert assessment on the results. All authors revised the paper.



*Competing interests.* The authors declare that they have no conflict of interest.

*Acknowledgements.* The authors would like to thank Sylvain Pujol, Nicolas Barry and Damien Lilas from CACG for providing information about the Gimone watershed, as well as measurements of stream flow and withdrawals. They also thank Vincent Bustillo (Cesbio) for
providing measurements of soil hydraulic conductivity, and Vincent Rivalland (Cesbio) for his help on modeling tools.



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
