# Peer review of "How to account for irrigation withdrawals in a watershed model"

_EGUsphere, 2023_

## Author Comment (AC1)

**Reply to RC1**

The present work details accounting for human activities i.e irrigation withdrawals and (maybe) dam releases in a hydrological model (SWAT+). It further compares two model set ups with two different crop phenological growth approaches i.e default approach using the heat units and another approach utilizing NDVI from remote sensing. The paper is well written and the diverse aspects crop representation, irrigation withdrawals, calibration approaches as well as, the adopted model setups are well explained and justified. Some limitations are also well recognized. To enhance the paper's significance and potential for inspiring future modeling efforts, it would benefit from providing additional information on the following points:

1. **Utilizing NDVI as a proxy for phenological development is a commendable approach, and it has been employed in previous studies as well as the Leaf Area Index from remote sensing (Ma et al., 2019; Chen et al., 2023). However, it is crucial to acknowledge the limitations associated with this method in the manuscript. For instance, it is important to note that NDVI may not be able to distinguish between different crops within the same field in cases of mixed cropping. Additionally, it can be challenging to differentiate between crops and other types of vegetation, such as shrubs. Consequently, while NDVI may be effective in agricultural catchments with predominantly single crops, its effectiveness might be limited in mixed or intercropping fields. Further elaboration and information regarding these limitations would be beneficial.**

   **Answer:** This is right. This methodology presented here based on Sentinel2 NDVI is well suited for plots with single crops plots. This question is challenging. And the first question is: does the spatial resolution of Sentinel2 images well adapted for mix-cropping?

   We suggest to add, line 405: « It should be noted that this method, which relies on NDVI signal at 10 m resolution and on single crop coefficients (Kc) per plot, is suitable for single crops but for mixed crop plots, further research is required ».

2. **The authors' preference for utilizing NDVI to derive phenology raises a question regarding the presence of cropping calendars from PKGC, as indicated in the "plant phenology" section. Nkwasa et al. (2022) have emphasized the recommendation of using crop calendars, which include plant and harvest dates. By using crop calendars, the authors show that the model can properly simulate the LAI in comparison to remote sensing LAI. It would be beneficial to have a discussion regarding the relative strengths of remote sensing over crop calendars. This discussion could highlight the advantages of remote sensing in situations where crop calendars are not available or feasible to use.**

   **Answer:** Intrinsically, having a crop calendar is optimal. Figure 4 shows that we have discrepancies between the emergences estimated from remote sensing images (NDVI) and those estimated from in-situ data (PKGC). However, as crop calendars are rarely available at territorial scale, using remote sensing to estimate phenology (emergence and flowering, harvest) remains a good compromise. Indeed, in our study, an "observed reconstructed emergence calendar" (from PKGC) is compared to a SWAT using NDVI, providing a better phenological representation than the Classical degrees-day SWAT approach. In the article pointed out by the reviewer, an "observed reconstructed planting calendar" from a global phenological dataset is used to force the model and the dataset for validation is the LAI detected by RS. Both studies seem to be consistent on the fact that forcing SWAT with "a reconstructed

calendar" or with RS data (NDVI) seems to improve the phenological representation into the model in comparison with a "classical" SWAT. One explanation for the imprecision of emergence dates estimated by remote sensing is due in particular to the presence of clouds at the time of crop emergence on our territory. One way of improving these inaccuracies is to use radar data, as explained in our conclusion.

We suggest to add, line 290: "Comparison between the PKGC database and the models clearly shows that spread emergence dates (SWAT-NDVI) fits better to real dates. This result seems consistent with results from the study of Nkwasa et al (2022) which have shown that using a crop calendar in the model fit better to the detection of LAI by RS. Precise crops calendar being rarely available at territorial scale, the use of vegetation indices by remote sensing seems then to offer a good compromise".

3. **Given the focus of this study on crop modeling, it would have been valuable to assess the model's performance in terms of crop yields. It raises the question of whether the two different implementations, utilizing the default approach with heat units versus the approach incorporating NDVI from remote sensing, would have a significant impact on crop yield estimates. This would enhance the understanding of their potential implications on agricultural assessments.**

**Answer:** Yield estimates are a complicated topic and would require a specific study. This was not the objective of our study, which focused on water consumption and hydrological processes. Our study aims first of all to evaluate the simulation of hydrological resources, and the impact of uptake water on water fluxes but not the impact of irrigation on crop production.

4. There is a lack of information regarding the representation of the two dams in the study. It is important to clarify whether this was achieved through the use of decision tables. If decision tables were employed, it would be beneficial to provide further elaboration, possibly in the supplementary material, to enhance the understanding of the dam representation method. Furthermore, it is crucial to elaborate how water was abstracted from the rivers in the model. Did this process involve the utilization of irrigation decision tables or water transfer tables? This information holds value for the scientific community, particularly in terms of reproducibility and ensuring the ability to replicate the study's methodology.

**Answer:** Dams are simulated via data series of outflow: i.e. the flow released from the dams is constraint and the model is forced at the dam's outlet to maintain a known flow rate in the river downstream. The location of the 2 gauge stations at the outlet of the dams can be seen on Figure 1-e. Irrigation is indeed controlled by decision tables (called decision "rules" in the paper). The main one, for corn, is displayed in the paper (Table2). The other values for other crop can be found into the Code files added into the supplementary material.

Water for irrigation of one HRU is abstracted from the closest river from this HRU, as explained in the first paragraph of section 2.5: "*The actual source of irrigation withdrawals (river or reservoirs) is not known. The PAR database however provides information about the distribution of authorized withdrawals between sources. We assumed real withdrawals distribution to be the same. It results in 58% of the water pumped from the rivers, whereas 42% comes from small reservoirs (36% from*

*connected reservoirs and 6% from disconnected ones). As small reservoirs (<10 ha) are not included in our setup, irrigation water was considered to come from the closest river from each HRU".*

No "water transfer table" were used here, because this option does not seem operational in version 60.5.3 of SWAT+. Instead, the source code was modified to reach our goal: (1) in the "action" routine, the channel is selected and irrigation is performed only if the channel contains enough water. (2) In the "sd_channel_control" routine, irrigation water is actually subtracted from the channel. The source code we used is available online (see "code availability" section).

We suggest to modify, line 114: "Stream flow coming out of the two dams was provided by the CACG" by "Measured stream flow coming out of the two dams was provided by the CACG and has been used as time series to force the discharge values into the model at the outlet of each dams"

**Editing and language comments**
**All comments with no specific answer will of be taken in consideration and modify in the manuscript**

Line 4: Do you mean "Two different crop models" or "Two different model setups of SWAT+"? "Two different model setups of SWAT+". Will be changed
Line 5: Which high resolution data from Sentinel-2?
We suggest to modify by *"high resolution NDVI data from Sentinel 2 level 2A product"*
Line 18: "accounts"
Line 20: "on spatial extents"
Line 23: "scenarios" – check through text.
Line 27: "forecasts"
Line 31: "users of the SWAT"
Line 39: "crop dynamics"
Line 44: "data have been"
Line 45: "and have bloomed"
Line 67: "decision rules with remote sensing"
Line 82: "irrigation has"
Line 86: What Sentinel-2 data is used? Please clarify and rephrase sentence. We suggest to modify by *"high resolution NDVI data from Sentinel 2 level 2A product"*
Line 356: "summer crops"
Line 399: What is an "unusual accuracy?"
we suggest to change by "satisfying accuracy"
Line 115: ……. high resolution remote sensing data of what? Please clarify on this and make clear throughout text. We suggest to modify by *"high resolution NDVI data from Sentinel 2 level 2A product"*
Line 119: What gap filled time series were created? Please clarify.
We suggest to modify by "… gap-filled NDVI time series."
Line 120: How long was the discharge data timeseries?
We suggest to modify by "The outlet discharge from 2012 to 2021 was also provided by the CACG…".
Line 150: Please clarify on what you mean by "true crop rotation"? Was this seasonal or annual? And what was the crop rotation (At least add this information to SI)? Also, this could be misleading I that it's interpreted as you are feeding the model with the actual plant and harvest dates for the different crops in the right rotations. Maybe clarify on this.

We suggest to replace « *the true crop rotation* » by « *annual crop rotation* »
and to add few more sentences for more information:
"*The most common rotations are « winter wheat-sunflower » (approx. 10% of fields), « winter wheat-winter barley-sunflower » (approx. 5%) and « corn monoculture » (approx. 2%). Most fields go through irregular rotations, that have all been included in the model*"

---

## Author Comment (AC2)

**Reply on RC2**

General comments:

This is an interesting modeling study performed in a context where many in situ observations are available (dam releases, river discharge, irrigation withdrawals, crop emergence dates). Two river discharge models including a representation of crop growth and irrigation are intercompared over a small watershed in southern France. One of the models is driven by satellite NDVI data.

The paper is reasonably well written but I think a separate Discussion section should be added.
**Answer:** Separating the discussion from the results would require a major restructuring of the document, which in our point of view would not bring any major improvement in understanding. Moreover, this request was not made by the first reviewer.

The Table and Figure captions are often incomplete.
**Answer:** Captions pointed out above on the "particular comments" will be modified.

Code availability is mentioned but nothing is said about in situ and satellite data availability. Are the various datasets used in this study available? Where?

**Satellite data from Sentinel-2:** we suggest to add, line 116:
"*These data were obtained through the Theia platform of the French Spatial Agency (https://theia.cnes.fr/atdistrib/rocket/#/home)*"

**Landuse data (rpg):** we suggest to add, line 112:
"…available for free on the French geographic institute (IGN - https://geoservices.ign.fr/rpg)"

**PKGC and Dams data**: we suggest to add, line 126:
"*PKGC and Dams management data were obtained through research agreements with the CACG and governmental services, and cannot be made public.*"

**Particular comments:**
**All comments with no specific answer will of be taken in consideration and modify in the manuscript**

- L. 42: A few more details should be given. For example, can emergence dates be determined from space? This is the point of references cited in the subsequent paragraph or example *"Rolle et al.2022 for sowing date"*
- L. 97: Unclear. Looks like you mean "winter sunflower". I would suggest adding another sentence explaining which crop is sown in winter.
We suggest to add "…sunflower during summer"
- L. 161: frpHU is undefined. Is this variable related to the growing degree-day concept?
We suggest to add line 163 "frPHU is the fraction of PHU from sowing date to day d"
- L. 173: I suggest replacing "Hnet" by "Rn".
- L. 174: wrong units. Replace kg m3 by kg m-3.

- L. 179: Is Zr constant in time? If not, what are the drivers of Zr?

We suggest to add line 179 "Zr is the root depth at day d".

- L. 187: What is the difference between ET0 and PET?

PET already considers the land use, but no stress coefficient. It is the ET that would occur if an unlimited amount of water was available in the soil.

In the 1st setup, PET is given in equation (3), and ET0 would be calculated with the same formula with a standard "rc" constant. In the 2nd setup, PET = (Kcb + Ke)*ET0

- L. 218: Are these parameters those listed in Table 3? Should be written here.

Yes, it is. We suggest to add a reference to table 3 at line 218

- L. 241: Caption of Fig. 2 is not complete. What are the boxplot percentile limits? What is the meaning of dots? Percentile limits are classically 25%/median/ 75%. Dot are the values used to plot the box. This information will be added to the caption

Why plotting a "relative change" for Ksat and Cn2? Why not using a logarithmic scale for example?

Because this is the classical approach to calibrate SWAT, the values of Ksat and Cn2 are different for each HRU so the calibration is performed by relative change in percent to keep the specificity of each HRU

- L. 242: 6 mm/year of change in storage. Does this mean that the soil is continuously storing more water?

This 6mm/year change in storage means that the SWC at December 31, 2021 (end of period) was 60mm (6mm/year * 10years) higher than the SWC at January 1, 2012 (beginning of period). We could say it means that the soil is stocking up water during the period. However, this is far from being a regular increase of the soil water content: the soil water content undergoes high and rapid variations of about 110 mm of amplitude (soil depth of 1m with AWC~1.1mm/mm) throughout the year, depending on the weather conditions.

- L. 249-251: description of BFI should be made in Section 2. Not here.

- L. 260: This should be indicated in the caption of Table 5.

- L. 261-264: the Moriasi classification should be described in Section 2. Not here.

- L. 270-274: move to a Discussion section.

- L. 289: why "-1"? That an error it will be removed

- L. 290-299: move to a Discussion section.

- L. 304-307: move to a Discussion section.

- L. 312: Figure caption is not complete. I can see 5 lines in these Figures: black, orange, light-orange, blue, light-blue. Only 3 are defined in the Figures, none in the Figure caption. I would replace "water meters" by "observations".

We suggest to add in the caption of figure 5: "*Light-orange and light blue zones represent the minimum and maximum values obtained with the 4 runs. It indicates the uncertainty of the simulation results.*"

- L. 321-322: Is SWAT-O more realistic than SWAT-NDVI for this time period?

- L. 323: Start a new section 4 (Discussion) here. End section 3 here.

Editorial comments:
- L. 18: accounts
- L. 20: extent
- L. 22: extent
- L. 44: have been
- L. 56: soil water balance calculation
- L. 81: take into account agricultural practices
- L. 94: the average yearly precipitation amount

- L. 103: delete the end of the sentence (unclear),
We suggest to modify" …and has little to do with hydrological processes" by "and hydrologic regime is not driven by natural processes"
- L. 109: French
- L. 166: when all the conditions
- L. 220: allowed
- L. 222: delete "indeed"
- L. 246: sensitive
- L. 295: lead
- L. 328: Figure 6 compares
- L. 348: 7-Up? Will be changed by *"upper part of table 7"*
- L. 376: approach.